# Can Creatine Supplementation Interfere with Muscle Strength and Fatigue in Brazilian National Level Paralympic Powerlifting?

**DOI:** 10.3390/nu12092492

**Published:** 2020-08-19

**Authors:** Carlos Rodrigo Soares Freitas Sampaio, Felipe J. Aidar, Alexandre R. P. Ferreira, Jymmys Lopes dos Santos, Anderson Carlos Marçal, Dihogo Gama de Matos, Raphael Fabrício de Souza, Osvaldo Costa Moreira, Ialuska Guerra, José Fernandes Filho, Lucas Soares Marcucci-Barbosa, Albená Nunes-Silva, Paulo Francisco de Almeida-Neto, Breno Guilherme Araújo Tinoco Cabral, Victor Machado Reis

**Affiliations:** 1Group of Studies and Research of Performance, Sport, Health and Paralympic Sports (GPEPS), Federal University of Sergipe (UFS), São Cristovão 49100-000, Sergipe, Brazil; rodrigosfsampaio@hotmail.com (C.R.S.F.S.); acmarcal@yahoo.com.br (A.C.M.); dihogogmc@hotmail.com (D.G.d.M.); raphaelctba20@hotmail.com (R.F.d.S.); 2Department of Physical Education, Federal University of Sergipe (UFS), São Cristovão 49100-000, Sergipe, Brazil; 3Program of Physical Education, Federal University of Sergipe (UFS), São Cristovão 49100-000, Sergipe, Brazil; 4Program of Physiological Science, Federal University of Sergipe (UFS), São Cristovão 49100-000, Sergipe, Brazil; 5College of Physical Education and Exercise Science, University of Brasília (UnB), Brasília 70910-900, Brazil; alexandreispf@gmail.com; 6Program in Biotechnology, Northeast Network in Biotechnology (RENORBIO), Federal University of Sergipe (UFS), São Cristovão 49100-000, Sergipe, Brazil; jymmyslopes@yahoo.com.br; 7Institute of Biological Sciences and Health, Federal University of Viçosa, Campus Florestal, Minas Gerais 35690-000, Brazil; ocostamoreira@gmail.com; 8Federal Institute of Education, Science and Technology of Ceará (IFCE), Campus of Juazeiro do Norte, Ceará 63040-540, Brazil; ialuskaguerra@gmail.com; 9Brazilian Paralympic Academy, Brazilian Paralympic Committee, São Paulo 04329-000, SP, Brazil; jffbepe@gmail.com; 10Laboratory of Inflammation and Exercise Immunology, Sports Center, Physical Education Scholl, Federal University of OuroPreto (UFOP), OuroPreto, Minas Gerais 35400-000, Brazil; lucasmarcucci@gmail.com (L.S.M.-B.); albenanunes@hotmail.com (A.N.-S.); 11Department of Physical Education, Federal University of Rio Grande do Norte (UFRN), Natal, Rio Grande do Norte 59078-970, Brazil; paulo220911@hotmail.com (P.F.d.A.-N.); brenotcabral@gmail.com (B.G.A.T.C.); 12Research Center in Sports Sciences, Health Sciences and Human Development (CIDESD), Trásos Montes and Alto Douro University, 5001-801 Vila Real, Portugal; victormachadoreis@gmail.com

**Keywords:** Paralympic powerlifting, supplementation, creatine, performance

## Abstract

The aim of the present study was to analyze the effect of creatine (Cr) supplementation on peak torque (PT) and fatigue rate in Paralympic weightlifting athletes. Eight Paralympic powerlifting athletes participated in the study, with 25.40 ± 3.30 years and 70.30 ± 12.15 kg. The measurements of muscle strength, fatigue index (FI), peak torque (PT), force (kgf), force (N), rate of force development (RFD), and time to maximum isometric force (time) were determined by a Musclelab load cell. The study was performed in a single-blind manner, with subjects conducting the experiments first with placebo supplementation and then, following a 7-day washout period, beginning the same protocol with creatine supplementation for 7 days. This sequence was chosen because of the lengthy washout of creatine. Regarding the comparison between conditions, Cr supplementation did not show effects on the variables of muscle force, peak torque, RFD, and time to maximum isometric force (*p* > 0.05). However, when comparing the results of the moments with the use of Cr and placebo, a difference was observed for the FI after seven days (U^3^: 1.12; 95% CI: (0.03, 2.27); *p* = 0.02); therefore, the FI was higher for placebo. Creatine supplementation has a positive effect on the performance of Paralympic powerlifting athletes, reducing fatigue index, and keeping the force levels as well as PT.

## 1. Introduction

Powerlifting (PL) is an international sport where competitors attempt to lift a maximum amount of weight in threeprimary lifts: the bench press, the squat, and the deadlift. These threelifts provide widely accepted measures of upper-body, lower-body, and total body strength [1,2,3]. At all levels of PL, each competing athlete is ranked based on the best of the threevalid attempts afforded for the bench press, squat, and deadlift [1,2,3]. The threeare then “totaled,” providing a measure of the total weight lifted and determining each athlete’s overall place in the competition. The skillful execution of each of the threeafforded lifts for the bench press, squat, and deadlift during competitions can be influenced by the subjects’ strength and muscle fatigue [1,2,3].

In this sense, many of these athletes have used ergogenic aids to keep body conditioning, enhancing recovery, and physiological adaptations during training programs and between competitions [3,4]. The efficacy of ergogenic has always attracted great attention, and numerous researchers have sought to combine ergogenic and exercise training programs to reinforce the benefits of training. Creatine (Cr) is a popular ergogenic aid among athletes at all levels [5,6]. Cr is a non-protein nitrogenous compound—methyl-guanidine-acetic acid—composed of three amino acids (arginine, glycine, and methionine). It is found mainly in skeletal muscle (95%) and plays an important role in rapid energy provision during muscle contraction through the ATP-PCr system [7].

Cr supplementation tends to potentiate the effect of strength training that would promote physiological responses and adaptations that positively interfere with the increase in muscle strength, power, hypertrophy, and local muscle endurance [8]. On the other hand, observing variables related to muscle recovery, there areindications that Cr supplementation could reduce muscle damage after exercise via sarcolemma stabilizing mechanisms [9] and regulate mitochondrial permeability [10]. Studies have demonstrated the beneficial effects of Cr supplementation on performance following resistance training [9,10,11].

Although the literature has shown the positive impacts of creatine supplementation [8,9,10,11], there is a lack of strong evidence about the efficacy of creatine supplementation on elite powerlifting athletes. More evidence is required to testing the efficacy of creatine to minimize fatigue index, which might enhance muscle strength, considering the need for new approaches that contribute to better performance.

Therefore, this was the first study to investigate the effects of creatine supplementation in elite Paralympic powerlifting athletes. We hypothesized that creatine might affect positively the muscle strength and reduce fatigue during high-intensity resistance training used in Paralympic powerlifting training. Thus, the aim of the present study was to analyze the effects of Cr supplementation on indicators of torque, force, time, and fatigue index in Paralympic powerlifting athletes.

## 2. Materials and Methods

### 2.1. Sample

The sample consisted of eight Paralympic powerlifting athletes participating in the extension project of the Federal University of Sergipe Brazil. All participants were Brazilian level competitors, eligible for the sport [12], and ranked among the ten best in their respective categories. Among the deficiencies, two athletes presented spinal cord injury due to accidents with an injury below the eighth thoracic vertebra; two with sequelae due to polio; two had lower limb malformation (arthrogryposis); two had cerebral palsy. The sampling power was calculated based on previous results of our studies [2,3], with an effect size of 0.98 that combined with a standard of α < 0.05 and β = 0.80. Thus, it was possible to estimate a sample power of 0.88, suggesting that the sample size has sufficient statistical strength to answer the research approach.

The characterization of the sample is shown in Table 1.

The athletes voluntarily participated in the study and signed a free and informed consent form in accordance with Resolution 466/2012 of the National Commission for Research Ethics (CONEP) of the National Health Council and the ethical principles of the latest version of the Declaration of Helsinki (and the World Medical Association). The project was submitted to the Research Ethics Committee of the Federal University of Sergipe and approved with the following opinion 2,637,882.

This study was carried out at the Federal University of Sergipe, from 09:00 h to 13:00 h, and was developed in four weeks, the first one aimed at familiarization and testing of 1 repetition maximum (1RM), force (force (Kgf) and force (N)), peak torque (PT), rate of force development (RFD), time to maximum isometric force (time), and fatigue index (FI), according to the items Section 2.4. Force Measurements and Section 2.5. Load Determination, respectively.

The experimental design of the study is provided in Figure 1.

### 2.2. Instruments

Weighing of the athletes was performed on a digital platform-type Michetti (Micheletti, São Paulo, SP, Brazil) electronic scale, with a maximum supported weight capacity of 3000 kg and a size of 1.50 × 1.50 m. For the bench press exercise, an official straight bench (Eleiko Sport AB, Halmstad, Sweden), approved by the International Paralympic Committee [12], with a total length of 210 cm was used. The IPC-approved powerlifting Olympic bar is serrated and has grooves in its material, has a total length of 220 cm, weighing 20 kg. On the bar, there is a marking for the narrowest and widest footprint, according to the International Paralympic Committee [12] official rules 2016–2017, ranging from 42 cm to 81 cm.

### 2.3. Supplementation

We chose to use a single-blind method with a treatment order that was not counterbalanced due to the lengthy washout time required for muscle creatine to return to pre-supplementation values [13]. Therefore, initially, participants ingested 20 g maltodextrin (placebo, Max Titanium^®^, Supley, Matão, SP, Brazil), followed by 7 days of washout period. Subsequently, 20 g of creatine monohydrate (Max Titanium^®^, Supley, Matão, SP, Brazil; 99.9% purity) was administered for another 7 days. The total daily amount of supplement was divided into four equal portions and consumed with food throughout the day. Creatine and placebo were identical in taste, color, texture, and appearance.

### 2.4. Force Measurements

Measurementsof muscle strength, fatigue index (FI), peak torque (PT), force (Kgf), force (N), rate of force development (RFD), and time to maximum isometric force (time) were determined by a Musclelab load cell (Model PFMA 3010e MuscleLab System; Ergotest, Langesund, Norway), attached to the adapted bench press, using 21 HN Simplex carabiners Spider HMS Simond (Simond, Chamonix, France), approved for climbing by *Union InternationaledesAssociations d’Alpinisme* (UIAA). A steel chain with a breaking load of 2300 kg was used to secure the load cell to the seat. The perpendicular distance between the load cell and joint center was determined and used to calculate joint torques and fatigue index [14].

Isometric peak torque (PT) was measured by the maximum torque generated by the upper limb muscles. The PT was determined by the product of the isometric force peak, measured between the load cell cable attachment point and the adapted bench press bench, which was adjusted so that an elbow angle was close to 90° and at a distance 15 cm from the starting point (chest to bar), verified with an apparatus for measuring the amplitude of angular movement, Model FL6010 (Sanny, São Bernardo do Campo, SP, Brazil). Participants were instructed to perform a single maximal movement, seeking elbow extension (as soon as possible) and relaxing for PT assessment.

For the fatigue index (FI) evaluation, the same exercise was performed, and the subjects determined to maintain the maximum contraction for 10 s, where the index was determined by dividing the initial PT in relation to the final PT, subtracted from one. FI = ((final PT-initial PT/final PT) × 100). Thus, the results in Newton (N) were conceived by the formula N = (M) × (C) × (H), where M = body mass in Kg, C = 9.81 (m·s^−2^), H = bar height relative to load (45.0 cm), corresponding to the height at which the equipment was fixed, adopting an angle of the forearm with the arm of 90°, adapted from the methodology from Milner-Brown et al. [15].

### 2.5. Load Determination

To determine the training load, the 1RM test was performed, on the adapted bench press [12], where each subject started the attempts with a weight that he believed could be lifted only once using the maximum effort. Weight increments were then added until the maximum load that could be lifted once was reached. If the practitioner could not perform a single repetition, 2.4 to 2.5% of the load used in the test was subtracted [16]. The subjects rested for 3–5 min between the attempts. The 1RM test was performed within two weeks at least 72 h prior to the intervention.

For the PT, force, RFD, time, and FI tests, three attempts were made in the PT test, where subjects were evaluated with the bar at 15 cm from the chest, and with an elbow angle of 90°, where they made the greatest force possible once, and this procedure was repeated three times after a rest of five minutes between the attempts. For the evaluation of the FI, the subjects remained to dothe maximum isometric contraction for one minute, with the bar 15 cm from the chest, and the loss of PT was verified between the 10 s and the initial moment of the test. The 10 s time was adopted in view of that shown in a study with Paralympic powerlifting [17], where the execution of 1RM, the target of the competition, would not amount to more than 10 s. All subjects underwent the test before and after training with a minimum interval of 10 min between the tests and the training session [18].

### 2.6. Intervention

The intervention protocol consisted of warm-up for upper limbs, using three exercises (abduction of the shoulders with dumbbells, elbow extension in the pulley, and rotation of the shoulders with dumbbells) with three sets of 10 to 20 repetitions [19]. Soon after, a specific warm-up was performed on the bench press with a 30% load of 1RM, 10 slow repetitions (3:1 s, eccentric:concentric), and 10 fast repetitions (1:1 s, eccentric:concentric),followed with five sets of bench press of five maximum repetitions (5 sets–85 at 90% RM with 3–5 min of rest), using a fixed load. During the test, athletes received verbal encouragement in order to achieve maximum performance [19]. To perform the bench press, an official straight bench (Eleiko Sport AB, Halmstad, Sweden), approved by the International Paralympic Committee, was used [12].

### 2.7. Statistic

The normality of the data was verified by the Shapiro Wilk and Z-score tests for asymmetry and kurtosis (−1.96 to 1.96). The assumption of normality was denied, and subsequently, the transformation of the data by the square root (i.e., from non-parametric to parametric) was not successful, and subsequently, the attempt to the logarithmic transformation of the data by the log on the basis of 10 was also unsuccessful. In this sense, comparisons between the medians of the same intervention (creatine × creatine; placebo × placebo) in the different conditions of the study (before, after training, after 7 days) were performed using the Kruskal–Wallis test. When differences were found, the Mann–Whitney U test was used to identify the different data set and, subsequently, Bonferroni correction. The differences between interventions (creatine × placebo) in the different conditions of the study (before, after training, after 7 days) were analyzed by the Mann–Whitney U test. The effect size between the median differences and their respective 95% confidence intervals was analyzed using the Cohen’s U³ index test, so the magnitude used was the one proposed by Espirito Santo and Daniel [20]: insignificant: <0.19; small: 0.20–0.49; average: 0.50–0.79; large: 0.80–1.29; very large: <1.30. All analyses were performed using open-source software R (version 3.6.2, R Foundation for Statistical Computing, Vienna, Austria), considering the significance of *p* < 0.05.

## 3. Results

Figure 2 shows the results of the effect of creatine supplementation (intra-conditions) in relation to the variables studied.

Regarding the use of creatine, Figure 2 shows that, for the RFD variable, significant differences were identified in the after training and after 7 days conditions in relation to the before condition (U^3^ = 1.33; CI 95%: [0.15]–[2.52]; *p* = 0.02), while for the use of placebo, there was no significant difference. Regarding thetime to maximum isometric force, there was a difference during the use of creatine for the after training condition compared to the before condition (U^3^ = 1.54; CI 95%: [0.32]–[2.76]; *p* = 0.01). In relation to the placebo, there was a significant difference in the conditions after training and after 7 days in relation to the before condition for the variable time to maximum isometric force (U^3^ = 0.76; CI 95%: [−0.34]–[1.87]; *p* = 0.04). There were significant differences in the after training condition in relation to the before and after 7 days conditions, and from the after 7 days condition to the before condition when using creatine for the fatigue index (U^3^ = 7.97; CI 95%: [4.76]–[11.1]; *p* = 0.0009). In relation to the use of placebo, significant differences were found in the conditions after training and after 7 days in relation to the before condition for the fatigue index (U^3^ = −12.9; CI 95%: [−17.9]–[−7.91]; *p* = 0.04).

Table 2 reports that the only statistical difference between interventions with the use of creatine and placebo was in the fatigue index (%) in the condition after 7 days (U^3^: 1.12; 95% CI: [−0.03]–[2.27]; *p* = 0.02), where the fatigue index (%) was higher for the intervention using the placebo.

Figure 3 shows graphically the behavior of the peak torque (Nm) and the percentage of the fatigue index (%) during the moments of the study (before, after training, and after 7 days), showing that in relation to the peak torque (Nm), the behavior was similar for creatine and placebo conditions. Whereas for the fatigue index (%), at the moment after 7 days, a lower percentage was demonstrated in the creatine condition in relation to the placebo condition.

## 4. Discussion

The objective of the present study was to analyze the effects of Cr supplementation on the indicators of torque, force, and muscle fatigue in athletes of Paralympic powerlifting. The main results were: (1) Cr supplementation did not show effects on the variables of muscle strength, peak torque, RFD, and time to maximum isometric force. (2) Cr supplementation reduced FIafter 7 days of use.

In the present study, when comparing Cr results with placebo, no significant differences were identified in relation to variables related to muscle strength. In this context, Zuniga et al. [21] examined the effects of 7 days of Cr supplementation on the strength of upper and lower limbs of 22 men and concluded that there was no statistically significant difference between the placebo condition and the Cr condition in all the muscle strength variables analyzed. In addition, Hamilton et al. [22] concluded that Cr supplementation combined with resistance training when relative loads and volumes were the same as a placebo condition did not result in a training advantage in absolute or relative strength performance. Syrotuik et al. [23] did not observe significant differences in the strength of the upper limbs when comparing two conditions after an intervention with Cr and placebo for 5 days.

Buts et al. [24] showed in a systematic review the scientific information from the years 1980 to 2017, stating that the data on the improvement of sports performance through creatinewere inconsistent. In addition, a meta-analysis made up of 100 different studies has shown that in the short term, Cr supplementation does not have significant effects on specific sports performance [25]. Moreover, in previous studies, Cr supplementation, in the short term, hasnot demonstratedeffects on specific strength levels related to different sports [26,27].

The results of the present research also showed that FIof the supplemented condition was approximately 16% lower when compared to placebo. In contradiction to the present study, Bazzucchiet et al. [28] evaluated 16 trained men with daily supplementation of 5 g of Cr + 15 g of maltodextrin or 20 g of maltodextrin and evaluated the maximum voluntary isometric contraction and dynamic contractions and fatigue for the flexor muscles of the elbow. The authors concluded that creatinecouldimprove neuromuscular function during voluntary contractions. In addition, they indicated that, according to the electromyography analysis, no significant differences were found between the conditions regarding muscle fatigue.

Possible explanations for the improvement of the Cr condition’s FI can be speculated, in addition to neuromuscular adaptation, by the increase in glycogen storage. An increase in glucose transporter type 4 (GLUT4) expression is suggested when Cr supplementation is combined with exercise [29]; positive effects of Cr supplementation are seen at initial levels and also by maintaining high levels of muscle glycogen for up to 2 h [30]. In addition, training tends to promote central and peripheral fatigue, along with other endocrine, immunological, inflammatory, and oxidative stress [31]. Moreover, training tends to modulate physiological adaptation and improve physical performance indicators [32]. In this context, supplementation can be used as a nutritional strategy for athletes to improve their physiological adaptation and performance [33].

Resistance to fatigue and the ability of the muscle to regenerate during intermittent high-intensity exercise are important qualities of neuromuscular function [34]. In addition, Cr can help protect against injury and muscle damage induced by strenuous contractile activities [35]. In athletes, who performed ultra-resistance tests, supplemented with 20 g/d of maltodextrin plus 50 g of Cr for 5 precompetitive days, decreased plasma creatine kinase activities, lactate dehydrogenase, preventing the increase in plasma oxaloacetic glutamic acid and glutamic pyruvic acid, activities have been observed [36].

In subjects with spinal cord injury, Cr levels improve muscle strength parameters, and this has a positive effect on the performance of daily activities and body health [37]. In Paralympic weightlifting athletes, who have suffered spinal cord injury, creatine can help to maximize the performance of the upper limbs by reducing the FI and providing a faster recovery during the practices provided by the sport [38].

In addition, it has been shown in the literature that creatine supplementation appears to reduce the spread of secondary injuries and improves the quality of the neuromotor system’s fitness [39]. However, caution is recommended regarding the water balance during the consumption of the supplement [24].

However, despite the relevance of the results, the present study had some limitations: (1) The evaluation was done in an acute way. (2) The improvement in FI might have been a result of the Cr supplementation time phase, that is, due to the fact of the time/order effect in which the study was carried out. Cr supplementation was performed in a second moment, and because of that, there might have been an adaptation to training, which might have influenced the athletes’ FI reduction. (3) Athletes’ diets were not changed during the study. Therefore, new studies should be carried out with a long washout period as well as other research designs.

## 5. Conclusions

It was concluded that creatine supplementation has a positive effect on the performance of elite Paralympic powerlifting athletes, reducing fatigue in the execution of the exercise, and keeping theforce levels.

## Figures and Tables

**Figure 1 nutrients-12-02492-f001:**
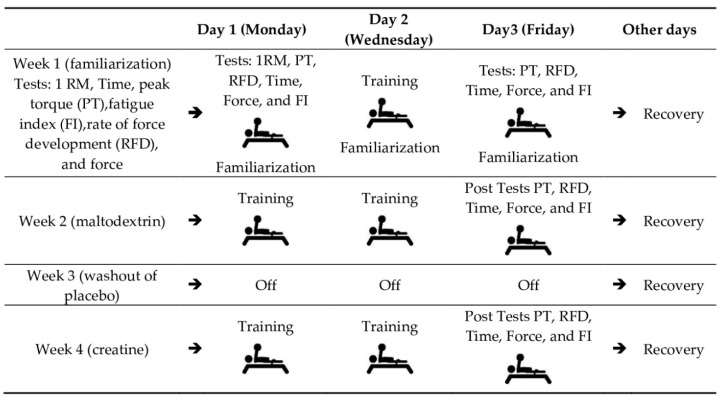
Experimental design—weekly schedule of tests and washout. 1RM: 1 repetition maximum. Training carried out three times a week, and the remaining days weredestined to rest [2].

**Figure 2 nutrients-12-02492-f002:**
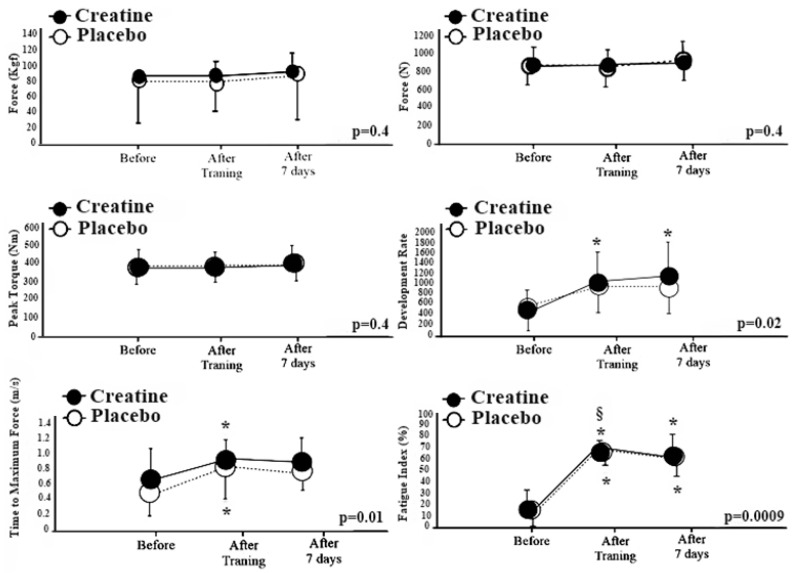
Intra-condition comparisons regarding force indicator at different times. Kgf = Kilograms force. N = Newtons. Nm = Nanometer. *p* = Value of the degree of statistical significance. m/s = Meters per Second. % = Percentage. * = Significant differences for the before condition. § = Significant difference for the after 7 days condition. *p* = Value of the degree of statistical significance.

**Figure 3 nutrients-12-02492-f003:**
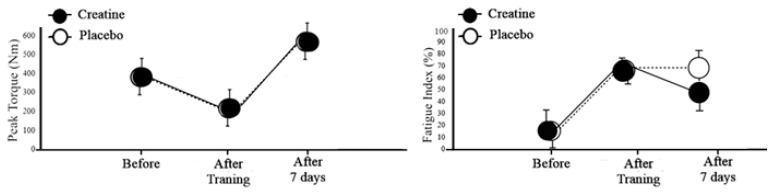
The behavior of peak torque (Nm) and fatigue index (%) at different times.

**Table 1 nutrients-12-02492-t001:** Characterization of subjects.

	(Mean ± SD)
Age (years)	25.40 ± 3.30
Weight (Kg)	70.30 ± 12.15
Experience (years)	2.45 ± 0.21
1RM Adapted Bench press (Kg)	119.99 ± 12.14 *
1RM/weight	1.71 ± 0.27 **

* All athletes with loads that keep them in the top 10 of their categories nationwide. ** Values above 1.4 on bench press would be considered elite athletes, according to Ball and Weidman. 1RM: 1 repetition maximum.

**Table 2 nutrients-12-02492-t002:** Comparison of moments using creatine and using placebo in the different conditions of the study.

	Before				After Training				After 7 Days			
Tests	Creatine		Placebo		Creatine		Placebo		Creatine		Placebo	
	MD	IIQ	MD	IIQ	MD	IIQ	MD	IIQ	MD	IIQ	MD	IIQ
Force (Kgf)	96.4	1.50	92.1	14.6	95.5	9.00	89.8	23.1	99.6	16.60	94.9	11.0
Force (N)	945.2	122.1	902.5	143.4	935.9	109.5	880.6	225.1	976.2	161.2	930.0	162.8
Peak torque (Nm)	425.3	54.9	406.1	64.5	421.1	48.9	396.2	101.6	439.3	73.30	418.5	73.4
Rate of force development	629.0	233.3	674.2	331.8	1.137	472.5	956.5	595.8	1.239	578.5	845.0	513.2
Time (m/s)	0.708	0.317	0.433	0.622	1.000	0.205	1.130	0.850	0.950	0.275	0.987	0.170
Fatigue index (%)	21.9	8.80	24.7	4.1	72.1	4.80	76.1	7.0	66.2	14.70	77.9 *	11.8

MD = Median; IIQ = Interquartile range. Kgf = Kilo grams force. N = Newtons. Nm = Nanometer. m/s = Meters per Second. % = Percentage, Time = Time to maximum isometric force. * = Significant statistical difference *p* = 0.02. *p* = Value of the degree of statistical significance.

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
