# Peer review of "Can Creatine Supplementation Interfere with Muscle Strength and Fatigue in Brazilian National Level Paralympic Powerlifting?"

_nutrients, 2020, doi:10.3390/nu12092492_

Round 1
Reviewer 1 Report
This manuscript by Carlos Sampaio and colleagues has examined the effect of 7 days of creatine supplementation on peak torque and fatigue rate among 8 paralympic weightlifting athletes using crossed-over design. The authors conclude that “creatine supplementation has a positive effect on the performance of paralympic powerlifting athletes, reducing fatigue index, increased force development rate and keep the strength levels”.
Although the study question is interesting, and novel for paralympic athletes, there are several studies that tested the effect of creatine supplementation on strength and fatigue among athletes.
Major comments:
The study design include 7 days of creatine monohydrate supplementation and 7 days of maltodextrin supplementation, with only one week of washout period between the cross-over. The authors should be aware that the washout period for muscle creatine to return to baseline following supplementation take up to 30 days, and may be longer for some individuals.
The study sample size is relatively low, with no justification in power or sample size analysis.
Miner comments:
Line 16, 20 - should be P=0.04.
In the abstract the authors conclude that creatine supplementation improved performance of paralympic powerlifting athletes, including FDR. However, the abstract results describe that both intervention and placebo “shown to be higher in relation to the previous condition” in terms of FDR (lines 14-16), without compression between groups.
Line 41 – Creatine should be changed to Cr.
The figures resolution need to be improved, and significant differences between groups/time points need to be added.
Line 148 – 5 sets * 5 reps, for how long did they rest? More details are needed.
Line 235 – The author mentioned the possibility that increase glycogen storage may improve neuromuscular function – when actually participant took the creatine with uncontrolled food.
Author Response
This manuscript by Carlos Sampaio and colleagues has examined the effect of 7 days of creatine supplementation on peak torque and fatigue rate among 8 paralympic weightlifting athletes using crossed-over design. The authors conclude that “creatine supplementation has a positive effect on the performance of paralympic powerlifting athletes, reducing fatigue index, increased force development rate and keep the strength levels”.
Although the study question is interesting, and novel for paralympic athletes, there are several studies that tested the effect of creatine supplementation on strength and fatigue among athletes.
Major comments:
The study design include 7 days of creatine monohydrate supplementation and 7 days of maltodextrin supplementation, with only one week of washout period between the cross-over. The authors should be aware that the washout period for muscle creatine to return to baseline following supplementation take up to 30 days, and may be longer for some individuals.
The sentence was adjusted and a reference was inserted for better clarification.
The study sample size is relatively low, with no justification in power or sample size analysis.
A paragraph was inserted justifying the sample size.
Miner comments:
Line 16, 20 - should be P=0.04.
It was changed as requested.
In the abstract the authors conclude that creatine supplementation improved performance of paralympic powerlifting athletes, including FDR. However, the abstract results describe that both intervention and placebo “shown to be higher in relation to the previous condition” in terms of FDR (lines 14-16), without compression between groups.
The abstract has been adjusted for a better understanding.
Line 41 – Creatine should be changed to Cr.
It was changed as requested.
The figures resolution need to be improved, and significant differences between groups/time points need to be added.
The figures have been adjusted for a better understanding.
Line 148 – 5 sets * 5 reps, for how long did they rest? More details are needed.
The information was entered as requested.
Line 235 – The author mentioned the possibility that increase glycogen storage may improve neuromuscular function – when actually participant took the creatine with uncontrolled food.
Although it was a speculation, we based on the article:
Hickner R, Dyck D, Sklar J, Hatley H, Byrd P. Effect of 28 days of creatine ingestion on muscle metabolism and performance of a simulated cycling road race. J Int Soc Sports Nutr. 2010;7:26.
In addition, a study limitation was inserted as a way to justify this sentence. The athletes' diet was not monitored during the study.
Reviewer 2 Report
General comments
The present study aimed to compare the Cr supplementation versus placebo on powerlifting performance parameters in elite Paralympic powerlifting athletes. Although this study is a very interesting topic in the field of sports nutrition, there are various problems as follows.
Introduction
- L28-35. In this paragraph, the explanation of powerlifting has been briefly carried out for the reader to understand. However, a paragraph was cited only by one prior study (Brown et al., 2013). This can cause plagiarism, so please change it.
- L50-53. In this section, the positive and negative aspects of performance change due to Cr supplementation were explained through previous studies. However, the authors should write to the PL athletes how to set the timing, amount, and timing of intake to maximize the effect of Cr supplementation based on previous research. This is very important to explain the necessity and justification of this study. Also, the explanation of Cr supplementation according to endurance activity is irrelevant to this study, so it is recommended to remove the content.
- In this study, the subjects were selected as Paralympic powerlifting athletes. If athletes with disabilities are selected for study, the reason for this should be explained in the introduction section.
Materials and Methods
- L65-66. There are many errors in grammar: As an example, change the ‘Time to Maximum Force (Time)’ to ‘time to maximum force (Time)’. Many of these errors are present throughout the sentence.
- In the description and explanation of Figure 1, English and Portuguese are used coexisting. Authors need a full fix to this in this manuscript.
- In material and methods section, first, it is necessary to describe the characteristics of the subject, and then, a detailed explanation of the research design with figure 1 is necessary. There is currently no study design section in manuscript, and no detailed explanation. Authors should add figures for the flow diagram of the consolidated standards of reporting trial and study design in the study design section.
- Authors must provide the IRB approval number and clinical trial number.
- There are many cases in which the period is incorrectly written as a comma in the text. This needs to be corrected. Ex> In Table 1, 25,43 ± 3,30. And authors need to add spaces between numbers and signs.
- L87-89. For all measuring equipment, the company name, city and country must be entered. Ex> a digital platform-type Michetti (Michetti, City, Brazil).
Results, Discussion, Conclusion
Did not the significant difference between treatments at each time point (Cr vs placebo) appear in all dependent variables? These results indicate that Cr supplementation has no effect on weightlifting performance unlike the hypothesis. Nevertheless, the authors claim that Cr supplementation has a positive effect on the performance of elite Paralympic Powerlifting athletes. And based on these positive conclusions, a discussion section was prepared. This is considered a very wrong approach. From the reviewer's point of view, Cr supplementation performed in this study did not have any positive effect as a result compared to placebo. The assumption of normality was rejected and the interaction effect was not verified based on repeated two-way ANOVA, authors can be explained that there is a Cr supplementation effect only when there is a significant difference between treatments at least at each time point. Based on this, the discussion and conclusion section should be rewritten. The same is true for abstracts. Additionally, authors must write the limitations section of this study.
Reference
The format of the previous study cited in the manuscript and reference section is incorrect. Make changes according to the MDPI and Nutrient journal forms.
Author Response
General comments
The present study aimed to compare the Cr supplementation versus placebo on powerlifting performance parameters in elite Paralympic powerlifting athletes. Although this study is a very interesting topic in the field of sports nutrition, there are various problems as follows.
Introduction
- L28-35. In this paragraph, the explanation of powerlifting has been briefly carried out for the reader to understand. However, a paragraph was cited only by one prior study (Brown et al., 2013). This can cause plagiarism, so please change it.
It was adjusted as requested and other references were cited
- L50-53. In this section, the positive and negative aspects of performance change due to Cr supplementation were explained through previous studies. However, the authors should write to the PL athletes how to set the timing, amount, and timing of intake to maximize the effect of Cr supplementation based on previous research. This is very important to explain the necessity and justification of this study. Also, the explanation of Cr supplementation according to endurance activity is irrelevant to this study, so it is recommended to remove the content.
The sentence was removed as requested.
On the other hand, there are no studies in the literature on PL and creatine supplementation that show how to set the timing, amount, and timing of intake.
In the fourth paragraph we mention this and reinforce this in the aim of this study.
- In this study, the subjects were selected as Paralympic powerlifting athletes. If athletes with disabilities are selected for study, the reason for this should be explained in the introduction section.
There is a description of these reasons on page 3 - lines 72-75. We agree with your placement. However, we believe that this sentiment should remain in the methodology.
Materials and Methods
- L65-66. There are many errors in grammar: As an example, change the ‘Time to Maximum Force (Time)’ to ‘time to maximum force (Time)’. Many of these errors are present throughout the sentence.
It was adjusted as requested.
- In the description and explanation of Figure 1, English and Portuguese are used coexisting. Authors need a full fix to this in this manuscript.
It was adjusted as requested.
- In material and methods section, first, it is necessary to describe the characteristics of the subject, and then, a detailed explanation of the research design with figure 1 is necessary. There is currently no study design section in manuscript, and no detailed explanation. Authors should add figures for the flow diagram of the consolidated standards of reporting trial and study design in the study design section.
It has been adjusted for a better understanding.
- Authors must provide the IRB approval number and clinical trial number.
This study was not submitted to Institutional Review Boards. However, all ethical aspects of the research were approved by the ethics committee of the Federal University of Sergipe - Brazil - CAAE: 2,637,882.
- There are many cases in which the period is incorrectly written as a comma in the text. This needs to be corrected. Ex> In Table 1, 25,43 ± 3,30. And authors need to add spaces between numbers and signs.
It was adjusted as requested.
- L87-89. For all measuring equipment, the company name, city and country must be entered. Ex> a digital platform-type Michetti (Michetti, City, Brazil).
It was adjusted as requested.
Results, Discussion, Conclusion
Did not the significant difference between treatments at each time point (Cr vs placebo) appear in all dependent variables? These results indicate that Cr supplementation has no effect on weightlifting performance unlike the hypothesis. Nevertheless, the authors claim that Cr supplementation has a positive effect on the performance of elite Paralympic Powerlifting athletes. And based on these positive conclusions, a discussion section was prepared. This is considered a very wrong approach. From the reviewer's point of view, Cr supplementation performed in this study did not have any positive effect as a result compared to placebo. The assumption of normality was rejected and the interaction effect was not verified based on repeated two-way ANOVA, authors can be explained that there is a Cr supplementation effect only when there is a significant difference between treatments at least at each time point. Based on this, the discussion and conclusion section should be rewritten. The same is true for abstracts. Additionally, authors must write the limitations section of this study.
The discussion was restructured and the study limitation was written according to what was requested.
Reference
The format of the previous study cited in the manuscript and reference section is incorrect. Make changes according to the MDPI and Nutrient journal forms.
The references were restructured.
Round 2
Reviewer 1 Report
The washout period in this study for the creatine supplementation was only 7 days, which is non-sufficient and remain the mejor concern. Washout period for muscle creatine to return to baseline should be more than 21 days.
Author Response
The washout period in this study for the creatine supplementation was only 7 days, which is non-sufficient and remain the major concern. Washout period for muscle creatine to return to baseline should be more than 21 days.
We understand your concern and inform you that we have restructured the writing of the methodology and abstract for a better understanding. In fact, as it was written, it was complex for a good understanding.
To clarify, the entire study was done initially with the placebo group and then, tests were performed with the group supplemented with creatine. Therefore, although it was 7 days of washout, this was for the placebo group.
Reviewer 2 Report
This study appears to have been well modified according to the reviewer's instructions.
Therefore, this reviewer approves this study in its current form.
Author Response
There was no request for adjustments